# A Prognostic Model to Predict Ruxolitinib Discontinuation and Death in Patients with Myelofibrosis

**DOI:** 10.3390/cancers15205027

**Published:** 2023-10-17

**Authors:** Francesca Palandri, Giuseppe A. Palumbo, Massimiliano Bonifacio, Elena M. Elli, Mario Tiribelli, Giuseppe Auteri, Malgorzata M. Trawinska, Nicola Polverelli, Giulia Benevolo, Alessia Tieghi, Fabrizio Cavalca, Giovanni Caocci, Eloise Beggiato, Gianni Binotto, Francesco Cavazzini, Maurizio Miglino, Costanza Bosi, Monica Crugnola, Monica Bocchia, Bruno Martino, Novella Pugliese, Marta Venturi, Alessandro Isidori, Daniele Cattaneo, Mauro Krampera, Fabrizio Pane, Daniela Cilloni, Gianpietro Semenzato, Roberto M. Lemoli, Antonio Cuneo, Elisabetta Abruzzese, Filippo Branzanti, Nicola Vianelli, Michele Cavo, Florian Heidel, Alessandra Iurlo, Massimo Breccia

**Affiliations:** 1IRCCS Azienda Ospedaliero-Universitaria di Bologna, Istituto di Ematologia “Seràgnoli”, 40138 Bologna, Italy; giuseppe.auteri2@unibo.it (G.A.); marta.venturi3@studio.unibo.it (M.V.); filippo.branzanti@aosp.bo.it (F.B.); nicola.vianelli@unibo.it (N.V.); michele.cavo@unibo.it (M.C.); 2Department of Scienze Mediche, Chirurgiche e Tecnologie Avanzate “G.F. Ingrassia”, University of Catania, 95124 Catania, Italy; palumbo.gam@gmail.com; 3Department of Engineering for Innovation Medicine, Section of Innovation Biomedicine, Hematology Area, University of Verona, 37129 Verona, Italy; massimiliano.bonifacio@univr.it (M.B.); mauro.krampera@univr.it (M.K.); 4Hematology Division, Fondazione IRCCS, San Gerardo dei Tintori, 20900 Monza, Italy; elena.elli@libero.it (E.M.E.); fabrizio.cavalca@gmail.com (F.C.); 5Division of Hematology and BMT, Azienda Sanitaria Universitaria Integrata di Udine, 33100 Udine, Italy; mario.tiribelli@uniud.it; 6Medicina Specialistica, Diagnostica e Sperimentale, Università di Bologna, 40126 Bologna, Italy; 7Division of Hematology, Sant’Eugenio Hospital, Tor Vergata University, 00133 Rome, Italy; trawinskamm@hotmail.com (M.M.T.); elisabetta.abruzzese@uniroma2.it (E.A.); 8Unit of Blood Diseases and Stem Cells Transplantation, Department of Clinical and Experimental Sciences, University of Brescia, ASST Spedali Civili of Brescia, 25121 Brescia, Italy; nicola.polverelli@gmail.com; 9Città della Salute e della Scienza Hospital, University Hematology Division, 10126 Torino, Italy; gbenevolo@cittadellasalute.to.it (G.B.); ebeggiato@cittadellasalute.to.it (E.B.); 10Department of Hematology, Azienda USL—IRCCS di Reggio Emilia, 42122 Reggio Emilia, Italy; alessia.tieghi@ausl.re.it; 11Hematology Unit, Department of Medical Sciences, University of Cagliari, 09124 Cagliari, Italy; giovanni.caocci@unica.it; 12Unit of Hematology and Clinical Immunology, University of Padova, 35122 Padova, Italy; gianni.binotto@unipd.it (G.B.); g.semenzato@unipd.it (G.S.); 13Division of Hematology, University of Ferrara, 44121 Ferrara, Italy; cvzfnc@unife.it (F.C.); antonio.cuneo@unife.it (A.C.); 14Clinic of Hematology, Department of Internal Medicine (DiMI), University of Genoa, 16126 Genova, Italy; maurizio.miglino@gmail.com (M.M.); roberto.lemoli@unige.it (R.M.L.); 15IRCCS Policlinico San Martino, 16132 Genova, Italy; 16Division of Haematology, AUSL di Piacenza, 29121 Piacenza, Italy; c.bosi@ausl.pc.it; 17Division of Hematology, Azienda Ospedaliero, Universitaria di Parma, 43126 Parma, Italy; mcrugnola@ao.pr.it; 18Hematology Unit, Azienda Ospedaliera Universitaria Senese, University of Siena, 53100 Siena, Italy; monica.bocchia@unisi.it; 19Division of Hematology, Azienda Ospedaliera ‘Bianchi Melacrino Morelli’, 89124 Reggio Calabria, Italy; brunmartin54@gmail.com; 20Department of Clinical Medicine and Surgery, Hematology Section, University of Naples “Federico II”, 80138 Naples, Italy; novypugliese@yahoo.it (N.P.); fabrizio.pane@unina.it (F.P.); 21Haematology and Haematopoietic Stem Cell Transplant Center, AORMN Hospital, 61100 Pesaro, Italy; alessandro.isidori@ospedalimarchenord.it; 22Hematology Division, Foundation IRCCS Ca’ Granda Ospedale Maggiore Policlinico, 20122 Milan, Italy; daniele.cattaneo@unimi.it (D.C.); alessandra.iurlo@policlinico.mi.it (A.I.); 23Department of Clinical and Biological Sciences, University of Turin, 10124 Turin, Italy; daniela.cilloni@unito.it; 24Internal Medicine II, Hematology and Oncology, Friedrich-Schiller-University Medical Center, 07747 Jena, Germany; florian.heidel@uni-greifswald.de; 25Department of Translational and Precision Medicine, Sapienza University, 00185 Rome, Italy; breccia@bce.uniroma1.it

**Keywords:** ruxolitinib, myelofibrosis, prognostic model

## Abstract

**Simple Summary:**

Despite significant clinical activity, 50% to 70% of patients discontinue ruxolitinib within 3 to 5 years. The identification of patients who are more likely to discontinue it early has now become of paramount practical importance, given the availability of new drugs that are either approved (i.e., fedratinib, pacritinib, and momelotinib) or undergoing advanced clinical investigation for patients with a suboptimal response or ruxolitinib resistance (i.e., navitoclax, pelabresib, and imetelstat). A retrospective, real-world analysis was performed on 889 MF patients treated with ruxolitinib from the observational “RUX-MF” Italian study. We investigated predictors of early ruxolitinib discontinuation and death on therapy in 889 MF patients. Results confirm that prolonged ruxolitinib administration is associated with improved OS when compared to earlier discontinuation. While this outcome may be expected at the 5-year timepoint, interestingly, the survival advantage was observed also categorizing patients at earlier timepoints.

**Abstract:**

Most patients with myelofibrosis (MF) discontinue ruxolitinib (JAK1/JAK2 inhibitor) in the first 5 years of therapy due to therapy failure. As the therapeutic possibilities of MF are expanding, it is critical to identify patients predisposed to early ruxolitinib monotherapy failure and worse outcomes. We investigated predictors of early ruxolitinib discontinuation and death on therapy in 889 patients included in the “RUX-MF” retrospective study. Overall, 172 patients were alive on ruxolitinib after ≥5 years (long-term ruxolitinib, LTR), 115 patients were alive but off ruxolitinib after ≥5 yrs (short-term RUX, STR), and 123 patients died while on ruxolitinib after <5 yrs (early death on ruxolitinib, EDR). The cumulative incidence of the blast phase was similar in LTR and STR patients (*p* = 0.08). Overall survival (OS) was significantly longer in LTR pts (*p* = 0.002). In multivariate analysis, PLT < 100 × 10^9^/L, Hb < 10 g/dL, primary MF, absence of spleen response at 3 months and ruxolitinib starting dose <10 mg BID were associated with higher probability of STR. Assigning one point to each significant variable, a prognostic model for STR (STR-PM) was built, and three groups were identified: low (score 0–1), intermediate (score 2), and high risk (score ≥ 3). The STR-PM may identify patients at higher risk of failure with ruxolitinib monotherapy who should be considered for alternative frontline strategies.

## 1. Introduction

Ruxolitinib was the first JAK1/2 inhibitor approved for the treatment of patients with myelofibrosis (MF) with splenomegaly and/or symptoms. In large prospective and retrospective cohorts, ruxolitinib has been shown to reduce splenomegaly and MF-associated symptoms in around 50% of the patients and improve quality of life in most patients, regardless of response. In addition, the introduction of ruxolitinib into clinical practice has been associated with a significant increase in life expectancy [1,2]. This beneficial effect was magnified in patients who achieved a spleen response, particularly if this occurred within the first 6 months of therapy. Indeed, in a pooled analysis of the COMFORT trials, a 5 dL increase from baseline in spleen volume correlated with a worse outcome; also, in the ruxolitinib phase 1/2 trial, MF patients with ≥50% reduction in spleen length were found to have significantly better survival [3,4]. Finally, we have previously shown that MF patients who did not achieve a stable spleen response to ruxolitinib by 6 months had significantly worse overall and progression-free survival compared to stable and unstable responders [5]. However, the durability of spleen response is limited, with more than 50% of responding patients losing such response within 3 to 5 years of therapy. Additionally, treatment-related toxicities, including on-target anemia and thrombocytopenia, infections, and second cancers, can substantially compromise the beneficial effect of ruxolitinib. As a result, despite significant initial clinical activity in many cases, 50% to 70% of patients discontinued ruxolitinib within 3 to 5 years [6,7]. After ruxolitinib discontinuation, the therapeutic possibilities have so far been very scarce, and the outcome is very poor, with median survival ranging between 13 and 23 months [8,9].

Identifying patients who are more likely to respond well to ruxolitinib is a major goal of clinical research because it would also have important implications in clinical practice. Indeed, it would also guide the choice of firstline therapy. We previously showed that patients with higher disease burden (large splenomegaly, low platelet count, and transfusion dependency), higher risk category, and a longer time (>2 years) interval between diagnosis and start of therapy have a lower probability of spleen response at 6 months [3]. Recently, the response to ruxolitinib after 6 months (RR6) model has been developed [4]. The RR6 model is based on four parameters, including RUX dose <20 mg twice daily at baseline, months 3 and 6; palpable spleen length reduction from baseline ≤30% at months 3 and 6; red blood cell (RBC) transfusions at months 3 and/or 6; RBC transfusions at all time points. This model identifies a high-risk category of patients with impaired survival who might benefit from a prompt treatment shift.

Ruxolitinib has been the only JAK1/JAK2 inhibitor available in clinical practice for over a decade. Recently, new-generation JAK inhibitors have expanded treatment options for patients with splenomegaly and/or symptoms. Fedratinib is a selective JAK2/FLT3/BRMD4 inhibitor approved by the FDA in 2019 based on the results of the JAKARTA and JAKARTA-2 studies that included ruxolitinib naïve and experienced patients, respectively [10,11]. After ruxolitinib failure, 55% and 21% of the patients achieved spleen and symptom response, respectively, at 6 months. Comparably to ruxolitinib, fedratinib may induce on-target hematological toxicity, with grade 3-4 anemia and thrombocytopenia occurring in 38% and 22% of patients, respectively [11]. Pacritinib is a JAK1-sparing inhibitor of JAK2/FLT3/IRAK1/activin A receptor type 1 (ACVR1) that has received accelerated FDA approval for patients with splenomegaly and/or symptoms and severe thrombocytopenia (platelet counts <50 × 10^9^/L) [12]. In addition to spleen and symptom improvement in patients with MF, results from retrospective analyses of subpopulations from the PERSIST-2 trial suggest pacritinib may also provide anemia benefits [13]. The JAK1/JAK2/ACVR1 inhibitor momelotinib has been studied in three phase 3 trials [14,15,16]. Across these trials, momelotinib showed consistent spleen, symptom, and anemia benefits for patients with MF.

The addition of drugs with a different mechanism of action than JAK inhibition to ruxolitinib monotherapy is currently under clinical investigation in order to improve responses and possibly minimize hematological toxicity. Investigational clinical trials comparing combination therapy with ruxolitinib plus new agents and ruxolitinib alone in the frontline setting are also ongoing.

Overall, the identification of patients who are more likely to discontinue ruxolitinib early has now become of paramount practical importance, given the availability of new drugs that are either approved (i.e., fedratinib, pacritinib, and momelotinib) or undergoing advanced clinical investigation for patients with suboptimal response or ruxolitinib resistance (i.e., navitoclax, pelabresib, and imetelstat) [17,18,19].

The aims of this study were to investigate predictors of early ruxolitinib discontinuation and death on therapy in 889 MF patients and to create a prognostic score which could help to identify those patients at higher risk of failure of ruxolitinib monotherapy.

## 2. Materials and Methods

### 2.1. Study Setting and Definitions

After IRB approval, the “RUX-MF” retrospective study collected data from 889 chronic phase MF patients who received ruxolitinib outside clinical trials in 26 hematology centers. All centers were asked to report, in an electronic case report form (e-CRF), their consecutive MF patients who received ruxolitinib according to standard clinical practice. The total number of medical files was reported by each center by data input into an electronic database developed to record all study data after the de-identification of the patients with an alphanumeric code to protect personal privacy. Any treatment decision, including starting ruxolitinib doses and dose adjustments over time, was at the physician’s discretion, based on patients’ characteristics and independent from participation in this study. After the first data entry, the follow-up information was validated with a revision of clinical data and specific queries were addressed to the participating center in case of inconsistent data. All patients were followed from 2013 until death or to data cut-off (28 June 2022), with a median follow-up time of 4.4 years.

This analysis focused on 410 patients, including 172 patients (19.3% of the total 889 cohort) who were alive on ruxolitinib after ≥5 years from RUX start (long-term ruxolitinib, LTR), 115 (12.9%) who were alive off ruxolitinib after ≥5 years (short-term ruxolitinib, STR), and 123 (13.8%) who died while on ruxolitinib after <5 years (early death on ruxolitinib, EDR) (Appendix A).

Risk category was assessed according to the Dynamic International Prognostic Score System (DIPSS) in primary MF (PMF) and to Myelofibrosis Secondary to Polycythemia Vera and Essential Thrombocythemia Prognostic Model (MYSEC-PM) in secondary MF (SMF) [20,21]. MF-related symptoms were assessed using the 10-item Myeloproliferative Neoplasm Symptom Assessment Form Total Symptom Score (MPN10-TSS). Spleen and symptoms responses were assessed by palpation and by routine MPN-TSS evaluation, respectively, according to 2013 IWG-MRT/European LeukemiaNet (ELN) criteria [22]. Progression to blast phase (BP) was defined according to WHO criteria [23].

### 2.2. Ethical Aspects

The RUX-MF study was performed in accordance with the guidelines of the institutional review boards of the participating centers and the standards of the Helsinki Declaration. All patients provided written informed consent. The promoter of this study was the IRCCS Azienda Ospedaliero-Universitaria S. Orsola-Malpighi, Bologna, which obtained approval from the Area Vasta Emilia Centro Ethics Committee (approval file number: 048/2022/Oss/AOUBo). The study was also approved by the local ethics committee of all participating centers (protocol code: RUX-MF). The study had no commercial support. 

### 2.3. Statistical Analysis

Statistical analysis was carried out at the biostatistics laboratory of the MPN Unit at the Institute of Hematology “L. and A. Seràgnoli”, IRCCS Azienda Ospedaliero-Universitaria, Bologna. 

Continuous variables were summarized by their median and range, and categorical variables were summarized by the count and relative frequency (%) of each category. Comparisons of quantitative variables between groups were carried out by a two-sample Wilcoxon rank-sum (Mann–Whitney) test; association between categorical variables was tested by the χ^2^ test. 

The cumulative incidence of progression to the blast phase was calculated, treating death as a competing event. Comparison of overall survival (OS) was carried out with the Cox regression model in multivariable analysis with DIPSS or MYSEC-PM, with adjustment for delayed entry and evaluation of the model’s performance in terms of goodness of fit.

To assess predictors of ruxolitinib discontinuation, the following baseline variables, selected on the basis of clinical plausibility, were explored using a Cox proportional hazards model: (1) age ≥ 65 years; (2) male sex; (3) platelet count < 100 × 10^9^/L; (4) hemoglobin < 10 g/dL; (5) transfusion dependence; (6) ruxolitinib starting dose < 10 mg twice daily; (7) total symptoms score (TSS) ≥ 20; (8) high or intermediate-2 DIPSS or MYSEC-PM; (9) diagnosis of primary MF; (10) bone marrow fibrosis grade ≥ 2 (in PMF patients only); (11) spleen palpable at more than 10 cm below the left costal margin; (12) interval between the start of ruxolitinib therapy and MF diagnosis longer than 2 years. Additionally, the absence of spleen and symptoms response at 3 months were evaluated.

Regressors associated with ruxolitinib discontinuation with *p* <0.05 in univariate analysis were jointly tested in a multivariate prognostic model. A rounded weight was associated with each risk factor based on its hazard ratio (HR). Sums of scores recognizing patients with a similar probability of early discontinuation were unified in risk categories.

Statistical analyses were performed using STATA Software, 15.1 (StataCorp LP, College Station, TX, USA).

## 3. Results

### 3.1. Study Cohort

Overall, 172 patients (19.3% of the total 889 cohort) survived on ruxolitinib after ≥5 years from RUX start and were defined as “long-term ruxolitinib” (LTR) patients. Conversely, 115 patients were alive but had discontinued ruxolitinib within 5 years from therapy start and were defined as “short-term ruxolitinib” (STR) patients. Compared to STR patients, LTR more frequently had a secondary MF (57% vs. 43%; *p* = 0.02), lower incidence of thrombocytopenia (platelets <100 × 10^9^/L) (3.5% vs. 10.0%; *p* = 0.03), anemia (hemoglobin <10 g/dL) (20.4% vs. 38.3%; *p* < 0.001), and transfusion dependence (10.6% vs. 38.3%; *p* = 0.003). 

One hundred and twenty-three patients died within 5 years from ruxolitinib start while on therapy and were defined as “early death on ruxolitinib” patients (EDR). EDR patients were older (median age: 73.7 years; *p* < 0.001), more frequently males (68.3%; *p* = 0.005), had higher DIPSS or MYSEC-PM (“High” disease-specific risk score: 21.3%; *p* < 0.001), higher leukocyte count (22%; *p* = 0.02), and transfusion dependence (34.8%; *p* = 0.003) compared to LTR and STR patients (Table 1).

Rates of spleen responses (SR) were non-statistically significant in LTR and STR patients at 3 months (35.2% vs. 24.2%, *p* = 0.07) and 6 months (39.5% vs. 32.6%, *p* = 0.29). Conversely, EDR patients had significantly lower rates of SR (18.4% vs. 35.2% in LTR and 24.2% in STR, *p* = 0.009 at 3 months and 17.5% vs. 39.5% and 32.6% at 6 months, *p* = 0.001). 

Rates of symptom responses (SyR) were non-statistically significant in LTR and STR patients at 3 months (72.1% vs. 61.0%; *p* = 0.06) and 6 months (75.8% vs. 72.8%; *p* = 0.6). Concerning EDR patients, SyR rates were significantly lower when compared with LTRs both at 3 and 6 months (*p* = 0.002 and *p* = 0.019, respectively), while they were non-statistically significant compared to STRs (*p* = 0.26 at 3 months, *p* = 0.1 at 6 months).

Rates of treatment-emergent any-grade anemia at 3 months were 64.2%, 67.8%, and 53.8% in LTR, STR, and EDR, respectively (*p* = 0.15), while at 6 months, they were 49%, 49.4%, and 33.9% (*p* = 0.09). In addition, even the rates of treatment-emergent thrombocytopenia were non-statistically significant in the three groups at 3 months (20.6%, 23.1%, and 29.7% in LTR, STR, and EDR, respectively; *p* = 0.21). At 6 months, LTR rates were significantly lower than STR (28.7% vs. 44%) (*p* = 0.008).

### 3.2. Outcome According to Timing of Ruxolitinib Discontinuation

At last contact, 285 out of 410 (69.5%) patients discontinued ruxolitinib, 14 (8.7%) progressed to a blast phase, and 226 (55.1%) died. Concerning treatment discontinuations, the main reasons were hematological toxicity (17.9%), lack/loss of spleen response (23.3%), and leukemic transformation (13.8%).

Cumulative incidence of the blast phase with death as a competing event was similar in LTR and STR patients (2.4% and 4.8% at 5 years, *p* = 0.08).

Overall survival (OS) was significantly longer in LTR patients (*p* = 0.002) (Figure 1). Specifically, the median OS was 84.5 and 63.0 months in LTR and STR patients, respectively. 

### 3.3. The Short-Term Ruxolitinib Prognostic Model (STR-PM)

In univariate analysis, baseline platelet count <100 × 10^9^/L (HR [95% CI]: 2.39 [1.28–4.45]; *p* = 0.006), haemoglobin <10 g/dL (HR [95% CI]: 1.99 [1.37–2.91]; *p* < 0.001), transfusion dependence (HR [95% CI]: 2.05 [1.32–3.18]; *p* = 0.001), ruxolitinib starting dose <10 mg twice daily (HR [95% CI]: 2.63 [1.72–4.00]; *p* < 0.001), total symptoms score ≥20 (HR [95% CI]: 1.49 [1.01–2.19]; *p* = 0.03), PMF diagnosis (HR [95% CI]: 1.59 [1.1–2.30]; *p* = 0.01), start of ruxolitinib after diagnosis >2 years (HR [95% CI]: 1.51 [1.05–2.18]; *p* = 0.03), and absence of spleen response at 3 months (HR [95% CI]: 1.57 [1.00–2.50]; *p* = 0.05) were associated with a higher probability of discontinuation of ruxolitinib within 5 years from the start of therapy.

In multivariate analysis, baseline platelet count <100 × 10^9^/L (HR [95% CI]: 2.02 [1.05–3.92]; *p* = 0.04), hemoglobin <10 g/dL (HR [95% CI]: 1.64 [1.07–2.52]; *p* = 0.02), PMF diagnosis (HR [95% CI]: 1.96 [1.28–2.99]; *p* = 0.002), absence of spleen response at 3 months (HR [95% CI]: 1.64 [1.02–2.61]; *p* = 0.04), and ruxolitinib starting dose <10 mg twice daily (HR [95% CI]: 1.95 [1.22–3.12]; *p* = 0.005) remained associated with a higher probability of discontinuation (Appendix A). 

Assigning one point to each significant variable in multivariate analysis, a prognostic model for short-term ruxolitinib (STR-PM) was built, and three groups were identified: low (score 0–1; 48.1% of the patients, n = 138), intermediate (score 2; 36.2% of the patients, n = 104), and high risk (score ≥ 3; 15.7% of the patients, n = 45), with a 5-year STR probability of 37.6%, 58.2%, and 90.4%, respectively (*p* = 0.001) (Figure 2). 

### 3.4. Expanding the STR-PM to Earlier Timepoints

An additional analysis was performed, looking at different timepoints of ruxolitinib discontinuation. 

Overall, 477 patients (53.7% of the total 889 cohort) were alive after ≥3 years from ruxolitinib start: 348 (73%) were on ruxolitinib (and were defined as long-term ruxolitinib 3 years, LTR 3 years) and 129 (27%) were alive but off ruxolitinib (and were defined as short-term ruxolitinib 3 years, STR 3 years). 

Additionally, 759 patients were alive after ≥1 year from ruxolitinib start: 636 (83.8%) were on ruxolitinib (LTR 1-year) and 123 (16.2%) were off ruxolitinib (STR 1-year). 

First, the association between LTR/STR status and overall survival was investigated.

Notably, OS improvements were also observed in patients who continued ruxolitinib over patients who discontinued the therapy within 3 years (Figure 3A), with a median OS of 64.7 and 54.0 months (*p* < 0.001). Accordingly, survival probability was higher for patients alive and on ruxolitinib after 1 year from therapy start (LTR-1 year and within 1 year, with a median OS of 45.9 and 32.3 months (*p* < 0.001)) (Figure 3B).

Second, the STR-PM was applied to both cohorts of LTR/STR 3 years and 1 year. Notably, the STR-PM was also applied considering earlier timepoints for ruxolitinib discontinuation.

In the STR-PM 3 years, patients were stratified as low (n = 192, 46.8% of the population), intermediate (n = 142, 34.7%), and high (n = 76, 18.5%) risk. The probability of early discontinuation at 3 years was 23.9%, 33.9%, and 48.1% in low, intermediate, and high-risk patients (*p* = 0.002) (Figure 4A).

In the STR-PM 1-year, the probability of early discontinuation was 12.6%, 18.0%, and 26.8% in low, intermediate, and high-risk patients (accounting for 45.2%, 34.6%, and 20.2% of the population, respectively) (*p* = 0.001) (Figure 4B).

## 4. Discussion

The therapeutic scenario of MF has undergone rapid changes in the last five years, thanks to the introduction of new *JAK2* inhibitors that are already or about to be available in clinical practice (e.g., fedratinib, pacritinib, and momelotinib) [13,24,25,26]. Furthermore, multiple studies are being conducted to assess both the efficacy and safety of new drugs with different mechanisms of action, either as monotherapy or in combination with ruxolitinib. Therefore, the identification of patients who are more likely to respond poorly to ruxolitinib monotherapy or to discontinue early the drug is becoming crucial in guiding clinical practice.

This study confirms that prolonged ruxolitinib administration is associated with an improvement of the OS when compared to earlier discontinuation [6,27]. While this outcome may be expected at the 5-year timepoint, interestingly, the survival advantage was observed also categorizing patients at earlier timepoints (namely, 1 and 3 years from ruxolitinib start). This advantage is unrelated to decreased leukemic progression, as the incidence of the blast phase was similar in patients who discontinued ruxolitinib and patients who are continuing it. This finding further suggests the absence of the significant disease-modifying activity of ruxolitinib monotherapy [28].

It is acknowledged that the outcome after ruxolitinib treatment is poor, regardless of the reason for discontinuation. Additionally, we have recently shown that the OS of patients who discontinued ruxolitinib in the chronic phase varied according to the type of salvage treatment after it, with patients receiving ruxolitinib rechallenge or novel agents having significantly improved survival compared to patients allocated to standard treatments (i.e., hydroxyurea, danazol, and splenectomy) [8,9]. To date, data on the impact of second-generation JAK2 inhibitors on the survival of patients after ruxolitinib failure are scant.

The probability of remaining on ruxolitinib may be predicted by the STR-PM, which consists of several factors, including ruxolitinib starting dose ≥10 mg BID, baseline hemoglobin ≥10 g/dL, baseline platelet count ≥100 × 10^9^/L, diagnosis of post-Polycythemia Vera/Essential Thrombocythemia MF, and achievement of a spleen response at 3 months. 

Notably, the presence of cytopenia is closely correlated with a higher probability of early ruxolitinib discontinuation and death in the STR-PM score. The association between the cytopenic phenotype and a worse outcome has been repeatedly observed and is likely to be related to multiple factors. Particularly, cytopenic MF is characterized by molecular alterations that have been found to correlate with worse outcomes, including low JAK2 variant allele frequencies, triple negativity, and the presence of high-risk subclonal mutations. In addition, cytopenic MF represents a great therapeutic challenge, with reduced options and a lower probability of clinical benefit that is related to the difficulty of administering adequate ruxolitinib doses and to a higher propensity to develop drug-related toxicities [17,18,19,29].

We recognize the limitations of this study, in particular its retrospective nature and the need to validate our findings in additional cohorts. In addition, immortalization bias may play a role when analyzing (conditional after 5 years of follow-up) survival in patients with longer and shorter duration of treatment. Finally, competing risks other than death, but probably leading to subsequent death (i.e., disease progression), may exist. Nonetheless, these results have been generated within a cohort of very well-characterized MF patients, followed in dedicated hematology centers, and data have been thoroughly verified after the first collection.

Additionally, these findings, although entirely innovative, are aligned with many previous clinical observations from real-world and prospective studies that have indicated cytopenia, low ruxolitinib dose, and absence of a response as markers for poorer outcomes of MF patients [3,4,8,30].

## 5. Conclusions

The STR-PM suggests, in particular, that the optimization of ruxolitinib dose right from the start of therapy and an early evaluation of the spleen response may be crucial to minimize the risk of early discontinuation and to promptly intercept patients that are more likely to fail ruxolitinib. Accordingly, low baseline ruxolitinib doses and the absence of a spleen response at 6 months are negative predictors of survival in the RR6 model [4]. 

While the RR6 score aims to identify those patients treated with ruxolitinib who merit a change in therapy early, the STR-PM aims to capture baseline characteristics associated with treatment failure. These features, combined with the failure to achieve a spleen response at 3 months, may identify patients who are less likely to respond to ruxolitinib monotherapy and should, therefore, be considered for alternative frontline strategies. Finally, higher DIPSS or MYSEC-PM risk and anemia are significantly associated with early death on ruxolitinib. Patients with these baseline characteristics may also require a personalized approach beyond ruxolitinib monotherapy.

The most important clinical value of this score is in the early recognition of patients who are at risk of suboptimal response to therapy and who would be candidates to prepare for enrollment in clinical studies, etc., upon start of ruxolitinib treatment. The score is also interesting from the pharmaco-economic point of view since it may allow a better drug allocation.

Whether new frontline therapies may improve outcomes in patients belonging to the worse STR-PM categories remains to be clarified. Comparably, whether and to what extent early switch in suboptimal responders may impact survival expectations and which second-line therapies should be selected are also to be determined in future studies.

## Figures and Tables

**Figure 1 cancers-15-05027-f001:**
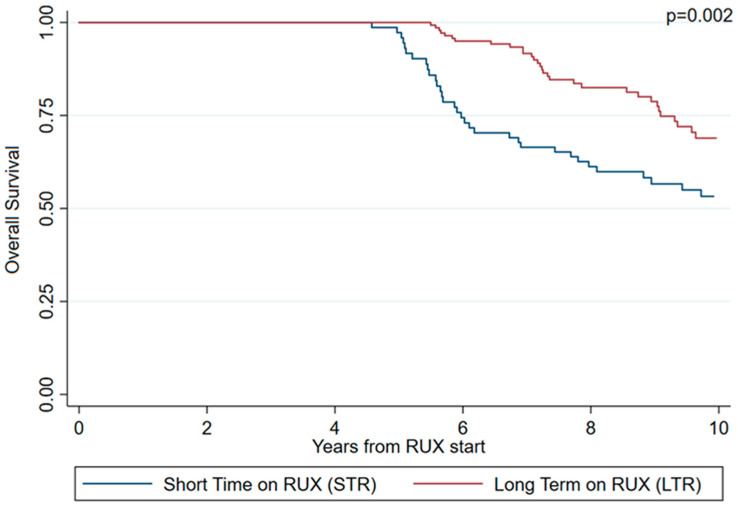
Overall survival (OS) adjusted for delayed entry according to STR/LTR phenotype at 5 years.

**Figure 2 cancers-15-05027-f002:**
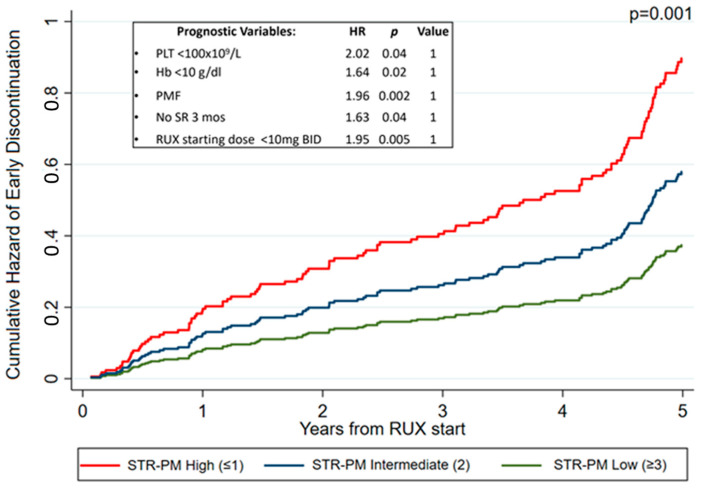
Probability of early ruxolitinib discontinuation based on the STR-PM considering patients who discontinued at 5 years.

**Figure 3 cancers-15-05027-f003:**
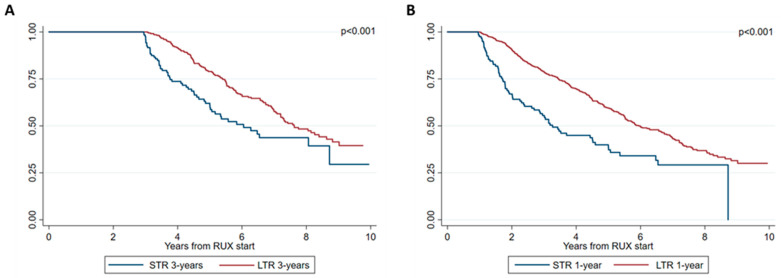
Overall survival (OS) adjusted for delayed entry according to STR/LTR phenotype at 3 years (**A**) and 1 year (**B**).

**Figure 4 cancers-15-05027-f004:**
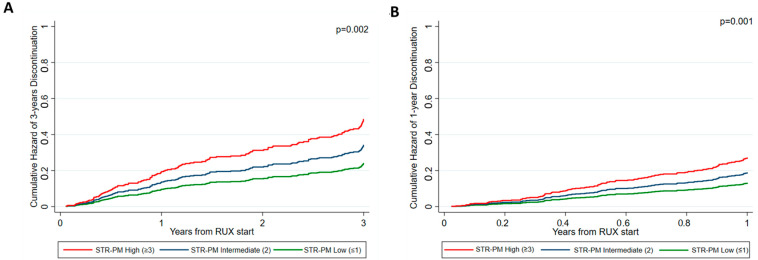
Probability of early ruxolitinib discontinuation based on the STR-PM considering patients who discontinued at 3 years (**A**) and 1 year (**B**).

**Table 1 cancers-15-05027-t001:** Characteristics at ruxolitinib start and spleen responses over time, according to early death on therapy (EDR), long-term (≥5 yrs) treatment with ruxolitinib (LTR), and short-term (<5 yrs) treatment with ruxolitinib (STR).

Characteristics	Total Study Cohort (n.410)					
	Follow-Up ≥ 5 Years			Follow-Up < 5 Years		
	Long-Term on RUX (LTR) n = 172 (42%)	Short-Term on RUX (STR) n = 115 (28%)	*p*	Early Death on RUX (EDR) n = 123 (30%)	*p* (LTR vs. EDR)	*p* (STR vs. EDR)
Age, median (range) Age ≥ 65, n. (%)	65.9 (26.5–85.8) 94 (54.7%)	65.4 (39.9–86.1) 59 (51.3%)	0.34 0.58	73.7 (38.4–86.3) 108 (87.8%)	**<0.001** **<0.001**	**<0.001** **<0.001**
Male sex, n. (%)	86 (50%)	60 (52.2%)	0.72	84 (68.3%)	**0.002**	**0.01**
Disease-specific risk score, n. (%) Int-1 Int-2 High	120 (69.8%) 44 (25.6%) 8 (4.6%)	70 (60.9%) 40 (34.8%) 5 (4.3%)	0.27	34 (27.4%) 62 (50.8%) 26 (21.3%)	**<0.001**	**<0.001**
PMF, n. (%)	74 (43%)	66 (57.4%)	**0.02**	68 (55.3%)	**0.04**	0.74
WBC, median (range) WBC > 25 × 10^9^/L, n. (%)	11.2 (2.7–70) 26 (15.1%)	10.4 (2.2–80) 13 (11.3%)	0.16 0.34	13.4 (2.2–92.3) 27 (22%)	**0.05** 0.13	**0.007** **0.03**
PLT, median (range) PLT < 100 × 10^9^/L, n. (%)	291 (32–1425) 6 (3.5%)	257 (60–1084) 11 (10%)	**0.02** **0.03**	250.5 (25–1887) 14 (11.5%)	0.12 **0.007**	0.54 0.63
Hb, median (range) Hb < 10 g/dL, n. (%)	12.2 (7–16.8) 35 (20.4%)	10.6 (5.7–16.3) 44 (38.3%)	**<0.001** **<0.001**	9.9 (5–16.7) 65 (53.3%)	**<0.001** **<0.001**	**0.006** **0.02**
Transfusion dependence, n (%)	17 (10.6%)	44 (38.3%)	**0.003**	41/118 (34.8%)	**<0.001**	**0.05**
Blasts, median (range) Blasts ≥ 1%, n. (%)	0 (0–10) 53 (31%)	0 (0–6) 44 (38.9%)	0.48 0.17	0 (0–10) 47/121 (38.8%)	0.25 0.16	0.64 0.99
Palpable spleen, median (range) Spleen ≥ 10 cm, n. (%)	10 (0–35) 84 (48.8%)	11 (0–31) 59 (51.3%)	0.12 0.58	10 (0–31) 60 (49.2%)	0.57 0.95	0.27 0.64
TSS, median (range) TSS ≥ 20, n. (%)	20 (0–70) 83/165 (50.3%)	20 (0–90) 69/110 (62.7%)	**0.005** **0.04**	23 (0–77) 83/115 (72.2%)	**<0.001** **<0.001**	0.25 0.13
Median years from MF diagnosis (range), Time from MF diagnosis >2 year	0.80 (0–22.35) 63 (36.6%)	2 (0–31.7) 57 (49.6%)	**0.01** **0.03**	1 (0–18.2) 50 (40.7%)	0.32 0.48	0.09 0.17
Ruxolitinib starting dose, n. (%) <10 mg BID	15 (8.7%)	29 (25.2%)	**<0.001**	25 (20.3%)	**0.004**	0.37
Spleen response, % on evaluable At 3 months At 6 months	35.2% 39.5%	24.2% 32.6%	0.07 0.29	18.4% 17.5%	**0.003** **<0.001**	**0.05** **0.02**
Symptoms response, % on evaluable At 3 months At 6 months	72.1% 75.8%	61.0% 72.8%	0.06 0.6	53.3% 61.5%	**0.002** **0.02**	0.26 0.10
Treatment-emergent anemia, % on evaluable At 3 months At 6 months	64.2% 49%	67.8% 49.4%	0.58 0.96	53.8% 33.9%	0.13 **0.04**	0.07 0.06
Treatment-emergent thrombocytopenia, % on evaluable At 3 months At 6 months	20.6% 28.7%	23.1% 44%	0.63 **0.008**	29.7% 39.2%	0.08 0.07	0.27 0.48

## Data Availability

The authors confirm that the data supporting the findings of this study are available within the article.

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
