# Peer review of "A Prognostic Model to Predict Ruxolitinib Discontinuation and Death in Patients with Myelofibrosis"

_cancers, 2023, doi:10.3390/cancers15205027_

Round 1

Reviewer 1 Report

The authors analyzed a plethora of data from a large cohort of patients in Italy with myelofibrosis treated with the JAK1/JAK2-inhibitor ruxolitinib. Despite the avalanche of data and jungle of statistics they could work out the main relevant and significant parameters and establish a prognostic model pinpointing patients who may benefit from a change in therapy.

Specific Points of Criticism and Suggestions for Alterations:

(1)  Line 49:  Maybe in the Abstract already mentioning the action of ruxolinitib (in parenthesis): "... (a JAK1/JAK2-inhibitor)...".

(2)  Line 49:  Why do these patients commonly discontinue ruxolitinib therapy? This may be added in a short aside (for example as follows or similar):  " ... due to therapy failure." or "... suboptimal response." or "... resistance."

(3)  Lines 56-57:  Maybe in parenthesis:  "... (xxx months versus xxx months)".

(4)  Line 133: "... collected data from 889 chronic phase patients ...".

(5)  Supplemental Figure 1:  This figure needs a list defining the abbreviations.

(6)  Table 1:  Maybe "n = xxx" (instead of „n.xxx“). The p-values in the most right column could be indicated and also marked by an asteriks (but possibly this clutters the table too much which is already full with numbers).

(7)  Table 1:  Maybe "n = xxx" (instead of „n.xxx“). The p-values in the column furthest to the right could indeed be indicated and also marked by an asteriks (but possibly this clutters the table too much which is already full with numbers).

(8)  Conclusions:  Maybe one or two sentences regarding an "outlook":  where to go next? what has to be done next? (in basic science and/or clinically).

Author Response

Thank you for your revisions. Please see the attachment for details.

Reviewer 2 Report

I have read the paper on predictors of ruxolitinib discontinuation with great interest. Please consider my comments as minor points aimed to further improve understanding of provided dataset. 

1) There are nearly significant spleen and symptom response rates that are treated as comparable when presenting the results. I suggest to rephrase the manuscript to consider them "not-statistically significant" instead of "comparable" since the study is underpowered for these comparisons (especially when considering that significant differences were present in ruxolitinib starting dose and in proportion of more advanced disease features between subgroups, implying that differences in other parameters associated with those two may also exist). 

2) the manuscript would benefit from the overview of reasons for drug discontinuation, i. e. disease progression, toxicity, administrative reasons, etc. to better understand what was the driving force behind the endpoint. I find the real clinical value of this score in the early recognition of patients who are under the risk of suboptimal response to therapy and who would be candidates to prepare for enrollment in clinical studies, etc. upon start of ruxolinitib treatment. The score is also interesting from the pharmacoeconomic point of view and feasibilty of ruxolitinib treatment.

3) the manuscript would benefit from the overview and comment on the second-line therapies (other JAK inhibitors, HU, supportive care, etc.) and prognosis post discontinuation in two subgroups, to better understand whether prognostic differences between short and long term on-ruxo experience truly exist as suggested by the post-5-years survival differences.

4) immortalization bias probably plays a role in phenomena observed when analyzing (conditional post 5-years of follow-up) survival in patients with longer and shorter duration of treatment. Also competing risks other than death, but probably leading to subsequent death (like disease progression, etc.) may exist. This should be more critically approached in the limitations and conclusion sections to more clearly state that probably intrinsic disease features but not duration of ruxolitinib treatment per-se are responsible for improved survival associated with longer use of ruxolinitib treatment.

Author Response

(The authors gave the same response as above.)
